# Codonoblepharonteae, a New Major Lineage among Orthotrichoideae (Orthotrichaceae, Bryophyta)

**DOI:** 10.3390/plants11243557

**Published:** 2022-12-16

**Authors:** Pablo Aguado-Ramsay, Isabel Draper, Ricardo Garilleti, Maren Flagmeier, Francisco Lara

**Affiliations:** 1Departamento de Biología, Facultad de Ciencias, Universidad Autónoma de Madrid, 28049 Madrid, Spain; 2Centro de Investigación en Biodiversidad y Cambio Global, Universidad Autónoma de Madrid, 28049 Madrid, Spain; 3Departamento de Botánica y Geología, Facultad de Farmacia, Universidad de Valencia, 46100 Valencia, Spain

**Keywords:** *Codonoblepharon*, *ITS*2, phylogeny, *rps*4, Taxonomy, *trn*G, *trn*L-F, *Zygodon*, Zygodonteae

## Abstract

Orthotrichoideae aggregates epiphytic mosses widespread throughout temperate regions and high tropical mountains of the world. Recently, important advances have been made in elucidating its phylogenetic relationships and evolutionary patterns. Fourteen genera are currently recognized within the subfamily, which are spread over two main tribes: Orthotricheae, comprising Orthotrichinae and Lewinskyinae, and Zygodonteae. Despite the progress, some groups have received little attention, as is the case of genus *Codonoblepharon*. Recent studies have suggested that this genus may represent a separate lineage from Zygodonteae, in which it traditionally has been considered. Although, none of the studies were conclusive as they did not include a representative sampling of the *Codonoblepharon* species. This work aims to evaluate the taxonomic position of *Codonoblepharon* and its phylogenetic relationships within Orthotrichoideae. For this purpose, we present an updated phylogenetic tree based on four different loci, one belonging to the nuclear genome (*ITS*2) and the rest to the plastid genome (*rps*4, *trn*G and *trn*L-F). The phylogenetic reconstruction recovers all samples of *Codonoblepharon* in a monophyletic group, sister to the rest of the subfamily, constituting a lineage independent of the two currently recognized tribes. For this reason, we propose the new tribe Codonoblepharonteae to accommodate *Codonoblepharon*.

## 1. Introduction

Mosses are the most diversified lineage among Bryophytes [1,2]. Recent years have witnessed important advances in the elucidation of the phylogenetic relationships among the major lineages of mosses, e.g., order level and above [3,4,5]. However, much remains to be clarified at lower levels, which is a key issue to establish a robust molecular-based classification, especially in the case of larger families.

Orthotrichaceae Arn. is the second most speciose family of mosses, with an estimated 900 species [6,7]. Most are epiphytic taxa, both from tropical and temperate environments, although in each of these major climatic regions one of the two subfamilies of the group predominates. Macromitrioideae Broth. are cladocarpous mosses with almost exclusively tropical distribution, whereas Orthotrichoideae Broth. includes acrocarpous mosses that inhabit temperate regions of both hemispheres and high tropical mountains [8]. Orthotrichoideae is better known, both in terms of specific diversity, e.g., [9] and phylogenetically, e.g., [6,10,11]. Recently, Draper et al. [12,13] have provided new and more complete insights in the phylogenetic framework of the subfamily, besides some of the evolutionary patterns underlying its complexity. According to these works, Orthotrichoideae is composed of fourteen genera. Ten of them are grouped into Orthotricheae, a tribe that in turn integrates two well differentiated lineages recognized as subtribes: (i) Orthotrichinae, that includes *Orthotrichum* s.str., the most diversified genus; (ii) and Lewinskyinae, which includes *Lewinskya* and *Ulota*, the other two major genera of the tribe. The other four recognized genera are currently integrated into Zygodonteae [12], with *Zygodon* s.str. as the most species-rich genus of this tribe (Table 1).

The phylogenetic reconstruction obtained by Draper et al. [12] did not conclusively resolve the analysed representation of Zygodonteae, as it separated its components in two well-supported clades but for which no robust conclusions could be drawn about their sister relationships. The results showed that one of the genera, *Codonoblepharon*, could constitute a separate lineage from Zygodonteae. Nevertheless, the placing of that lineage in a polytomy, together with a clade containing the rest of the genera of Zygodonteae and a clade containing the genera of Orthotrichoideae, advised postponing the possible consequences for the systematic of the subfamily until obtaining robust evidence.

*Codonoblepharon*, a genus initially conceived as grouping very heterogeneous taxa [14], was later circumscribed by Goffinet and Vitt [8] to segregate a section from *Zygodon* (Sect. *Bryoides* Malta), mainly characterized by the absence of papillae in its leaf cells [15]. Thus conceived, *Codonoblepharon* contains about seven mainly tropical and southern hemisphere species [14,16], although there are discrepancies about the ascription of *C. forsteri* (Dicks.) Goffinet, the only one restricted to the northern hemisphere. While studies based on morphological or biogeographical evidence suggest the exclusion of this species from *Codonoblepharon* [8,14], others based on phylogenetic reconstructions with partial taxa representation [6,12] indicate that it should be recognized as part of this genus.

In the present work we evaluate the hypothesis, based on the results obtained by Draper et al. [12], that *Codonoblepharon* represents a lineage independent of Zygodonteae. Our main objective is to achieve a robust phylogenetic reconstruction that would unequivocally reflect the relationships of *Codonoblepharon* within the subfamily Orthotrichoideae. Besides, we pursue to obtain more information about the phylogenetic structure of *Codonoblepharon* and the taxonomic position of *C. forsteri*.

## 2. Results

The independent analysis of the four markers resulted in phylogenetic trees with congruent topologies, which allowed concatenation. As shown in Table 2, the concatenated matrix resulted in a total length of 2065 bp, of which 550 were parsimony informative. In addition, indel coding added 153 informative variable positions. All the analyses performed (indels treatment and phylogenetic reconstruction method) were congruent. The best resolved phylogeny was originated from the BI analysis of the concatenated matrix with coded indels (Figure 1).

Representatives of Orthotrichoideae were resolved in three distinct clades. Zygodonteae, which until now had been treated as a single group, is divided in two maximally supported (1/100) lineages. On the one hand, *Codonoblepharon* appears as a monophyletic group (1/100), which is resolved as a sister clade to the rest of the subfamily. On the other hand, a lineage constituted of *Pentastichella*, *Australoria* and *Zygodon* is differentiated (1/99). This indicates that Zygodonteae s.l. represents a paraphyletic group, as was traditionally conceived. Conversely, Orthotricheae is grouped in a monophyletic clade (1/100), containing Lewinskyinae (1/100) and Orthotrichinae (1/74).

All the molecular markers used recover the samples of *Codonoblepharon forsteri* in a clade with maximal support that is nested within *Codonoblepharon* and sister to *C. pungens* (Müll.Hal.) A.Jaeger. This group is in turn sister to *C. menziesii* Schwägr., the type species of this genus.

## 3. Discussion

This work presents an updated molecular phylogeny with representation of all the accepted genera of the subfamily Orthotrichoideae based on Draper et al. [12], except for the recently described *Rehubryum* F.Lara, Garilleti and Draper [13] from Lewinskyinae. Special attention was paid to the tribe Zygodonteae and the genus *Codonoblepharon*. The variability and large number of analyses that have been conducted, consistent with each other, minimize the possibility of topological inconsistencies. The only analysed marker that has shown ambiguous alignments due to its high variability is nuclear *ITS*2. Nevertheless, the exclusion of these ambiguities did not affect phylogenetic results.

Orthotrichoideae is a taxonomically complex subfamily, which has led to numerous supraspecific reassignments. The species of *Zygodon* s.l., including the group with smooth leaf cells now segregated in the genus *Codonoblepharon*, have long been considered a large and important natural group. Unlike *Orthotrichum* s.l., the other traditional large genus of acrocarpous Orthotrichaceae, *Zygodon* s.l., includes mostly dioicous mosses, with sporophytes bearing a long seta and usually producing vegetative propagules on variably long and branched filamentous supports arising from the stem, never directly from the leaves, although propagules may be grouped in the leaf axils. Other distinguishing characteristics are the growth of the colonies forming lax turfs, rarely mats or cushions, the relatively small leaves with non or slightly recurved margins and little or no cell differentiation along the lamina, and the cucullate non-plicate calyptra, typically devoid of hairs. They typically grow on old or decaying bark of large trees or stumps. As in other groups of Orthotrichaceae, several species are saxicolous, either facultative or, more rarely, obligate [15,16] and [17] (pp. 15–135).

Malta [15], in the only world monograph on the genus, recognized a total of 77 *Zygodon* species grouped into four sections. Most of them (ca. 86%) belonged to the globally distributed section *Euzygodon* Müll.Hal. Section *Stenomitrium* Mitt. included a single species, with Andean and Patagonian distribution and deviant morphology due to its robust and creeping stems, leaves in pentastichous arrangement and dimorphic basal leaf cells. Section *Obtusifolii* Malta also contained a single species characterized by lingulate leaves with a rounded apex and papillose calyptra and a wide disjunct distribution including populations in Southeast Asia, Australasia, South America and Mexico. Finally, section *Bryoides* Malta included nine species, mainly distributed in the southern hemisphere and, furthermore, occurring in the north in some tropical localities and in Europe, characterized by smooth leaf cells. Malta [15] argued that this latter section represented a natural group that might merit subgeneric rank, as smooth cells are unusual among Orthotrichaceae. Malta’s taxonomic delimitation was basically followed by Calabrese [16], even though Goffinet and Vitt [8] had reinstated a few years earlier the genus *Codonoblepharon* for most of the species of section *Bryoides* and *Bryomaltaea* Goffinet to segregate *Zygodon obtusifolius* Hook (Table 3).

*Zygodon* s.l. has been treated at various taxonomic ranks, including family (Zygodontaceae Schimp.) and subfamily (Zygodontoideae Broth.). These mosses have numerous characters shared with other Orthotrichaceae, but some deviate and more closely resemble representatives of other families, such as Ditrichaceae Limpr., Grimmiaceae Arn. or Pottiaceae Hampe [8,18,19], that are not phylogenetically close. In fact, Schimper [20] proposed the segregation of *Zygodon* as a family that he considered intermediate between Orthotrichaceae and Weissiaceae Schimp. (=Pottiaceae p.p.). Most of its components are currently included in Pottiaceae, and the proposal also included *Amphidium* Schimp., now placed in Amphidiaceae M.Stech. Among the characters that can be considered deviant are those that have been highlighted as being particularly significant [21]: the cucullate non-plicate calyptra and the orthotropic stems forming short turfs, not cushions or creeping mats as in other Orthotrichaceae. The current consideration of *Zygodon* s.l. as a tribe within Orthotrichoideae is relatively recent [8]. However, regardless of the taxonomic rank given to the group, it has traditionally been considered a fairly homogeneous entity, well differentiated from the rest of Orthotrichaceae. This idea that Zygodonteae is a natural and quite uniform group was only partially questioned in early molecular studies [6], which allowed the segregation of *Bryomaltaea obtusifolia* (Hook.) Goffinet as part of a phylogenetically distant lineage. However, this did not diminish the general consideration of Zygodonteae. Recently, however, Draper et al. [12] concluded that the genus *Zygodon* is a polyphyletic artificial group and their results supported the distinction of *Pentastichella* with the inclusion of *Pleurorthotrichum* Broth. and the establishment of the new genus *Australoria* (Table 1 and Table 3). Thus, Malta’s sections of *Zygodon* [15] are now treated as separate genera, *Zygodon* s.s. being currently restricted to the representatives of the section *Euzygodon* (Table 3).

The results of the present work support the distinction of *Codonoblepharon* as a separate lineage, as earlier suggested by Goffinet et al. [6] and Draper et al. [12]. Unequivocal evidence for this was obtained by inclusion in the analysis of a wide representation of the former Zygodonteae, including species with smooth leaf cells and a greater number of representatives with papillose leaf cells. The phylogenetic reconstruction obtained here places the monophyletic group constituted by *Codonoblepharon* as a sister group of the clade that includes all the other Orthotrichoideae. This suggests the need for recognition of this lineage as a separate tribe, which we propose to name Codonoblepharonteae; it is the third within the Orthotrichoideae, along with Zygodonteae and Orthotricheae. Thus, *Codonoblepharon* changes from being considered just a section of *Zygodon* to a major independent lineage among Orthotrichoideae. This taxonomic proposal is morphologically supported by smooth leaf cells, which is an exclusive character of Codonoblepharonteae within the subfamily. This classification is paralleled by Macromitrioideae, which contains a single genus characterized by entirely smooth leaf cells, *Schlotheimia* Brid., being also considered a separate tribe, Schlotheimieae Goffinet [8]. Future phylogenetic studies on Macromitrioideae may reveal the true relationships of this group with smooth cells within the subfamily.

The phylogenetic reconstruction (Figure 1) also shows that the lengths of the branches and the topology within Codonoblepharonteae are similar to what can be observed in other main lineages, such as Zygodonteae, which includes up to three genera. In contrast, in the new tribe all the terminals belong to a single genus, *Codonoblepharon*. This leads us to consider that the taxonomic diversity of the group is yet to be investigated and that the segregation of *Codonoblepharon* into several separate genera could be possible. Further in-depth studies are needed to unravel this possibly overlooked diversity, giving special attention to *C. pungens*, which, according to Malta [15], constitutes the systematic weak point of the group and to *C. minutum* (Müll.Hal. and Hampe) Matcham and O’Shea which in our reconstructions appears as the sister species of all the congeners included in the analysis. Further evidence of the possibly overlooked diversity among Codonoblepharonteae is the strong phylogenetic structure obtained for *C. menziesii* which could have considerable taxonomical and biogeographical significance. The New Zealand samples of *C. menziesii* are separated from the Australian and Californian ones, which could imply that they correspond to different taxa and might support the idea of Shevock [22] that the occurrence of *C. menziesii* in western North America is due to a recent introduction from Australia. Since Malta [15] already recognized a strong intraspecific morphological variability within *C. menziesii*, the current concept could hide a complex of species and calls for a deep integrative study.

*Codonoblepharon forsteri* is limited to Europe and northwestern Africa, making it the only representative of the genus with a Holarctic distribution [23]. Regarding its phylogenetic reconstruction, the samples are nested within the lineage of Codonoblepharonteae. Their segregation in a subclade together with *C. pungens* is significant as they are the only two autoicous species of the genus included in the analysis. In Orthotrichoideae, most genera are either dioicous or autoicous [12,13], which could support their segregation into a separate genus, although, once again, it is preferable to await the results of a more complete morphological and molecular study to resolve this in a robust and accurate manner.

## 4. Materials and Methods

### 4.1. Taxon Sampling

The final analyses were composed of 65 samples from 47 different taxa, out of which 94 sequences from 27 samples were newly obtained for this study. These include a representative from *Australoria* [*A. chilensis* (Calabrese and F.Lara) F.Lara, Garilleti and Draper], 4 taxa from *Codonoblepharon* [*C. forsteri*, *C. menziesii*, *C. menziesii* var. *angustifolium* (Malta) Matcham and O’Shea and *C. minutum*], the two known taxa from *Pentastichella* [*P. chilensis* (Broth.) F.Lara, Garilleti and Draper, and *P. pentasticha* (Mont.) Müll.Hal. ex Thér.], and 8 taxa from *Zygodon* [*Z. catarinoi* C.A.Garcia, F.Lara, Sérgio and Sim-Sim, *Z. fragilifolius* Broth. ex Malta, *Z. hookeri* var. *leptobolax* (Müll.Hal.) Calabrese, *Z. intermedius* Bruch and Schimp., *Z. rupestris* Schimp. ex Lorentz, *Z. seriatus* Thér. and Naveau, *Z. stirtonii* Schimp. and *Z. trichomitrius* Hook. and Wilson]. The information regarding these sequences is available in Appendix A.

The identification of those samples was based on the analysis of microscopic characters of the leaves, stems, propagules, rhizoids, seta and capsules, according to the descriptions and taxonomical criteria of Lewinsky, Matcham and O’Shea, and Calabrese and Lara et al. [14,16,24] and [17] (pp. 17–27).

### 4.2. DNA Isolation and Amplification

Only the tip of a single gametophyte shoot from each sample was selected for DNA extraction to prevent contamination. The rest of the gametophyte, and the sporophyte if present, were preserved in a microscope slide fixed with glycerogelatin to allow identification revision. DNA was extracted using the standard DNeasy Plant Mini Kit protocol (QIAGEN). Nucleotide sequences were amplified by PCR from four genomic regions (Table 4): one of them from to the nuclear genome (*ITS*2) and the other three from to the plastid genome (*rps*4, *trn*G and *trn*L-F). PCRs were performed using Ready-To-Go™ PCR Beads (Amersham Pharmacia Biotech Inc.) in a final volume of 25 μL, initially with 2 μL of DNA, and in the case of subsequent PCR failure, with up to 10 μL. Amplification protocols are specified in Table 5. Amplification’s success was verified by electrophoresis and PCR products were purified using Exol/FastAP (Thermo Fisher Scientific, Spain) with 1 μL of Exonuclease and 4 μL of FastAP enzymes per tube, applying 37 °C and 85 °C for 15 min each. Finally, cleaned PCR products were sequenced by Macrogen. Two reads were obtained for each product, which were aligned using Geneious 2022.0.2.

### 4.3. Molecular Analyses

The reconstruction of the phylogenetic relationships was conducted in a framework of the 65 aforementioned samples (Appendix A). The genus *Codonoblepharon* was represented by 19 samples from 5 of the 8 accepted taxa. Outgroup were composed of 44 samples from 40 different taxa, representing 13 of the 14 genera currently accepted in the family [12,13]. Special attention was paid to Zygodonteae. In addition, based on the results of Goffinet et al. [6], *Macrocoma lycopodioides* (Schwägr.) Vitt was used to root the tree as a representative of Macromitrioideae, which was accompanied by *Leratia obtusifolia* (Hook.) Goffinet.

A matrix was constructed for each marker using PhyDE-1 0.9971 [31]. In order to eliminate uncertainties, 3′ and 5′ ends were trimmed in each matrix. Specifically, 99 bp in 3′ and 8 bp in 5′ for *ITS*2, 57 bp in 3′ and 10 bp in 5′ for *rps*4, 37 bp in 3′ and 63 bp in 5′ for *trn*G and 4 bp in 3′ and 18 bp in 5′ for *trn*L-F. The trimmed matrices were automatically aligned with MAFFT (*Multiple Alignment using Fast Fourier Transform*) using the EMBL-EBI multiple sequence alignment service [32]. Indels can sometimes lead to ambiguous alignments. Thus, each marker was analysed separately in three different ways: (1) indels considered as missing information, (2) indels coded as informative with the simple method of Simmons and Ochoterena [33] implemented in SeqState [34] and (3) removing divergent alignment zones using Gblocks 0.91b [35,36]. Gblocks values were modified, setting minimum length of a block to 5, allowing gap positions to “with half”, minimum number of sequences for a flank position to 25 and maximum number of contiguous non conserved positions to 10. The best evolutionary model and partition scheme were selected with PartitionFinder 2.1.1 [37,38,39] using the Bayesian information criterion (BIC).

Phylogenetic analyses were performed with BI using MrBayes Windows version 3.2.7a [40,41,42] and with ML using RAxML 8.2.12 [43] implemented in CIPRES [44]. BI analyses were run for 10,000,000 generations, saving trees and parameters every 1000. The initial 25% of the samples were not included in the consensus tree. Variable and informative positions were quantified with MEGA 11 [45,46]. Initially, each marker was analysed separately to detect possible incongruences, although the congruence of these markers in this group of mosses had already been verified in other studies, e.g., [12]. Once the congruence between the different resulting trees was visually confirmed, a combined matrix with the four markers was generated. The resulting phylogenetic trees were visualized and edited with FigTree 1.4.4 [47].

## 5. Conclusions

The present study has confirmed that the genus *Codonoblepharon* constitutes a separate lineage, which is resolved as sister from both Zygodonteae and Orthotricheae. This justifies the recognition of this lineage as an independent tribe, which we propose to name Codonoblepharonteae. Additionally, our results reveal that the diversity of the group is yet to be known, and that future integrative studies are necessary.

## Figures and Tables

**Figure 1 plants-11-03557-f001:**
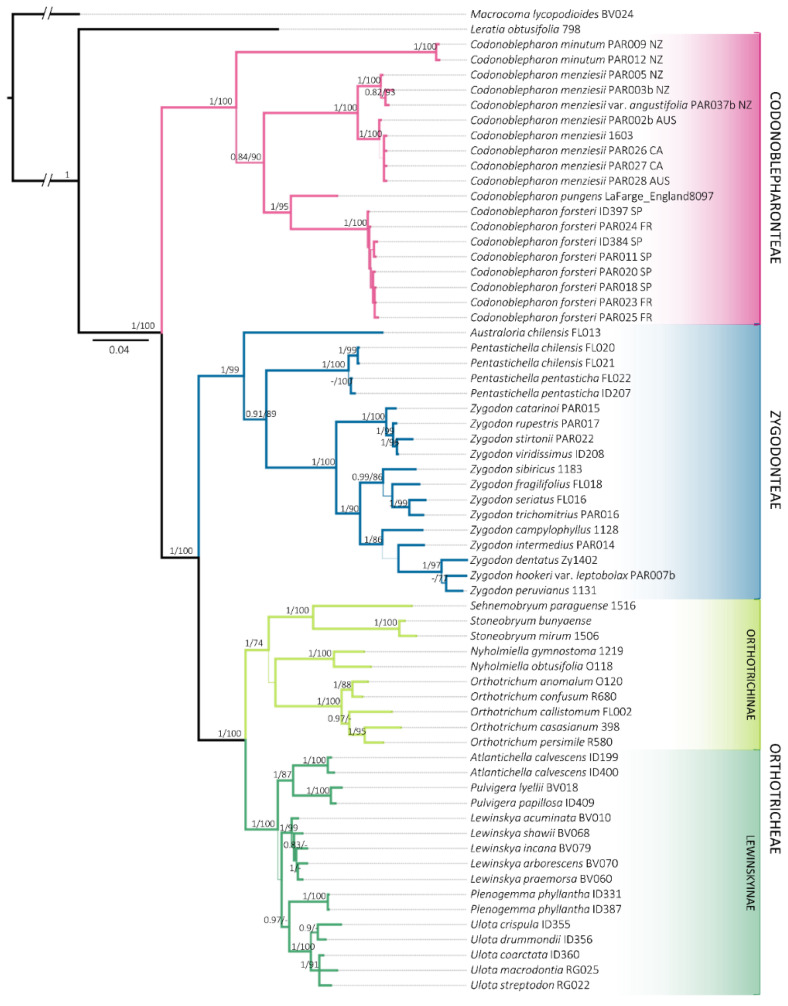
Consensus phylogenetic tree obtained from the Bayesian Inference analysis with the combined matrix (*ITS*2, *rps*4, *trn*G, *trn*L-F and indels coded). Posterior probabilities greater than 0.8 for BI and Bootstrap values greater than 70% for ML are included. Sequences information is available in Appendix A. The locality where the samples were collected is indicated for Codonoblepharonteae: Australia (AUS), California (CA), France (FR), New Zealand (NZ) and Spain (SP).

**Table 1 plants-11-03557-t001:** Classification of Orthotrichoideae according to Draper et al. [12,13].

Orthotrichoideae Broth.
Tribe **Orthotricheae** Engler
Subtribe **Orthotrichinae** F.Lara, Garilleti and Draper
***Orthotrichum*** Hedw.
***Sehnemobryum*** Lewinsky and Hedenäs
***Stoneobryum*** D.H.Norris and H.Rob.
***Nyholmiella*** Holmen and E.Warncke
Subtribe **Lewinskyinae** F.Lara, Garilleti and Draper
***Atlantichella*** F.Lara, Garilleti and Draper
***Lewinskya*** F.Lara, Garilleti and Goffinet
***Pulvigera*** Plášek, Sawicki and Ochyra
***Plenogemma*** Plášek, Sawicki and Ochyra
***Rehubryum*** F.Lara, Garilleti and Draper
***Ulota*** D.Mohr
Tribe **Zygodonteae** Engler
***Australoria*** F.Lara, Garilleti and Draper
***Codonoblepharon*** Schwägr.
***Pentastichella*** Müll.Hal. ex Thér.
***Zygodon*** Hook. and Taylor

**Table 2 plants-11-03557-t002:** Number of variable and informative positions for the concatenated matrix and for the four analysed markers.

		*ITS*2	*rps*4	*trn*G	*trn*L-F	Total
Ingroup	Variable sites	115	91	35	43	257
	Informative sites	37	57	11	38	143
	Indel informative sites	8	5	2	9	24
Total	Variable sites	319	195	146	163	820
	Informative sites	228	135	91	99	550
	Indel informative sites	81	15	28	29	153
Positions in data matrix		514	647	554	360	2065
Indels in data matrix		145	28	52	42	267
Evolutionary models		HKY + I + G	HKY + G	HKY + I + G	HKY + G	GTR + I + G

**Table 3 plants-11-03557-t003:** Evolution of the classification of *Zygodon* and related genera since Malta [15]. Genera not considered as belonging to tribe Zygodonteae in the respective study are marked with two asterisks (**).

Malta, 1926 [15]	Goffinet and Vitt, 1998 [8]	Calabrese, 2006 [16]	Goffinet et al., 2004 [6]	Draper et al., 2021 [12] and Current Proposal
*Zygodon* sect. *Euzygodon*	*Zygodon*	*Zygodon* subg. *Zygodon* sect. *Zygodon*	*Zygodon*	*Zygodon*
	*Leptodontiopsis*	*--*		
*Zygodon* sect. *Stenomitrium*	*Stenomitrium* ^1^	*Zygodon* subg. *Zygodon* sect. *Stenomitrium* ^4^	*Pentastichella*	*Pentastichella*
*–*	*Pleurorthotrichum* ^2^	*–*	*Pleurorthotrichum* ^2^	
*–*	*--*	*–*	*--*	*Australoria*
*Zygodon* sect. *Obtusifolii*	*Bryomaltaea*	*Zygodon* subg. *Obtusifolii*	*Leratia* **	*Leratia* **
*–*	*Leratia*	*–*		
*Zygodon* sect. *Bryoides*	*Codonoblepharon* ^3^	*Zygodon* subg. *Codonoblepharon*	*Codonoblepharon*	*Codonoblepharon* **

^1^ Nom. illeg. ^2^ A monotypic genus subsequently considered a synonym of *Pentastichella* by Draper et al. [12]. ^3^ Not including *Codonoblepharon forsteri*. ^4^ Including *Zygodon bartramioides* Dusén ex Malta and *Z. chilensis* Calabrese and F.Lara, both of them later considered in *Australoria* by Draper et al. [12].

**Table 4 plants-11-03557-t004:** Primers used for PCR amplification and sequencing.

Region	Primer name	Sequence	Authorship
*ITS*2	ITS2-F	CGGATATCTTGGCTCTTG	Ziolkowski and Sadowski [25]
	ITS2-R	CCGCTTAGTGATATGCTTA	
*rps*4	rpsA	ATGTCCCGTTATCGAGGACCT	Nadot et al. [26]
	trnaS	TACCGAGGGTTCGAATC	Souza-Chies et al. [27]
*trn*L-F	trnLc-104	TAAGCAATCCTGAGC	Vigalondo et al. [28]
	trnFF-425	CTCTGCTCTACCAACT	
*trn*G	trnGF-Leu	GGCTAAGGGTTATAGTCGGC	Werner et al. [29]
	trnGr	GCGGGTATAGTTTAGTGG	Pacak and Szweykowska-Kulinska [30]

**Table 5 plants-11-03557-t005:** PCR amplification protocols. Cycles are displayed in the first column of each region, heat and time in the second.

*ITS*2		*rps*4		*trn*G		*trn*L-F	
1	94 °C 1 min	1	94 °C 5 min	1	94 °C 5 min	1	95 °C 5 min
30	94 °C 1 min	30	95 °C 30 s	40	94 °C 30 s	38	94 °C 30 s
	59 °C 1 min		52 °C 1 min		52 °C 40 s		47 °C 1 min
	72 °C 1 min 30 s		68 °C 30 s		72 °C 1 min 30 s		72 °C 30 s
							94 °C 30 s
1	72 °C 5 min	1	68 °C 7 min	1	72 °C 8 min	1	72 °C 10 min

## Data Availability

Sequences used in this study were submitted to GenBank.

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
