# Peer review of "Codonoblepharonteae, a New Major Lineage among Orthotrichoideae (Orthotrichaceae, Bryophyta)"

_plants, 2022, doi:10.3390/plants11243557_

Round 1

Reviewer 1 Report

The manuscript deals with the infrafamiliar classification of the Orthotrichaceae - one of the most diverse and taxonomically difficult group of mosses. Reordering the tribe level classification originates from the robust molecular phylogenetic evidences and generally looks fine. The dataset allows well backuped taxonomic inferences both at the infra(sub-)familial and infrageneric levels.

Only one point in the analysis could need in improvement: as the authors pointed out, several parts of ITS2 are hard to align (so, two options were checked, with and without these parts, but the reader will found this in Discussion instead of M&M). Actually such regions indeed may be leaved as they are, but indels from such region hardly may be considered as reliable. So, in the future backbone phylogenetic studies I would suggest not including indels from this region. Then, several techniques like sampling nucleotide substitution model throughout the GTR model space (setting nst=mixed) could improve the topology, but seemingly there is nothing here to improve. Results might be improved by more clear description of the tree topology. At the moment the main point, that Zygodonteae s.str. form a maximally supported clade with Orthotricheae and Lewinskyeae (which suggest no other option than dividing Zygodonteae into two tribes) is not pointed out

I wonder a little why authors presented their taxonomic inferences in so shy manner, I would propose them in a clearer way (see comments in attached pdf). Discussion is perfect, although with the map of Codonoblepharon distribution it could look even better.

The language is a bit unusual for me and I made loads of correction in the pdf, but since I am not a native speaker, they are suggested just for author's consideration (although some points look rather crucial, please, pay attention to these)

Reviewer 2 Report

The manuscript entitled Codonoblepharonteae as a new major lineage among Orthotrichoideae (Orthotrichaceae, Bryophyta) has been submitted to the journal Plants (MDPI). 

I found the article well presented and provided important information to clarify the phylogenetic position of dubious taxa which has been confused and classified over the last decades. 

The methods and the relative analysis it is sound and appropriate. Results and Discussion are well presented. I do advise the Editor to accept the manuscript. 

However, I have a few minor comments that I would like to add. 

In Table 1 After Plenogemma there is an ‘R’ which I do not understand the meaning of, please clarify. 

The species considered and assessed in this study have a wide geographical distribution and the biogeographical approach has been discussed in minor detail only in the final part of the discussion. I was expecting a longer discussion. 

One aspect that I have not understood clearly is the division between European and Mediterranean distribution. 

When it is used the Mediterranean distribution does it includes the Mediterranean basin distribution therefore an Anfi-mediterranean distribution. or Mediterranean 'climatic' distribution, including also extra areas as California or Chile etc.

In theory, if you write it has a European distribution, for some regions imply also the Mediterranean basin, Spain, France, Italy, Greece etc. 

Whereas a continental European distribution or Atlantic European distribution brings more information on the biogeographical discussion. All these to say that the division between Mediterranean and European distribution is not clear for a non specialist and it would be appreciated if it is better explained in the discussion. 

Author Response

In Table 1 After Plenogemma there is an ‘R’ which I do not understand the meaning of, please clarify.

Response: Thank you very much for the indication, it is now deleted.

One aspect that I have not understood clearly is the division between European and Mediterranean distribution.

Response: To clarify this issue, it has been changed to European and northwestern African distribution.

We would like to thank you for your time and help to improve the initial version of this manuscript.

Reviewer 3 Report

An excellent paper with which the authors are to be congratulated. My suggestions for linguistic improvement are in the manuscript.

Author Response

The comments related to the writing have gladly been incorporated, and all proposed changes have been accepted in the manuscript. We also welcomed the proposed changes to the title and layout. We would like to thank you for your time and help to improve the initial version of this manuscript.
